# Genome Size, Chromosome Number and Morphological Data Reveal Unexpected Infraspecific Variability in *Festuca* (Poaceae)

**DOI:** 10.3390/genes12060906

**Published:** 2021-06-11

**Authors:** Gloria Martínez-Sagarra, Sílvia Castro, Lucie Mota, João Loureiro, Juan A. Devesa

**Affiliations:** 1Department of Botany, Ecology and Plant Physiology, Rabanales Campus, University of Cordoba, 14071 Cordoba, Spain; bv1dealj@uco.es; 2Centre for Functional Ecology, Department of Life Sciences, University of Coimbra, Calçada Martim de Freitas, 3000-456 Coimbra, Portugal; scastro@bot.uc.pt (S.C.); luciemota.bio@gmail.com (L.M.); jloureiro@bot.uc.pt (J.L.); 3Botanical Garden of the University of Coimbra, Calçada Martim de Freitas, 3000-456 Coimbra, Portugal

**Keywords:** *Festuca*, flow cytometry, genome size, nuclear DNA content, Poaceae, tetradecaploid

## Abstract

Polyploidy has played an important evolutionary role in the genus *Festuca* (Poaceae), and several ploidy levels (ranging from 2n = 2*x* = 14 to 2n = 12*x* = 84) have been detected to date. This study aimed to estimate the genome size and ploidy level of two subspecies belonging to the *F. yvesii* polyploid complex by flow cytometry and chromosome counting. The phenotypic variation of the cytotypes was also explored, based on herbarium material. The genome size of *F. yvesii* subsp. *lagascae* has been estimated for the first time. Nuclear 2C DNA content of *F. yvesii* subsp. *summilusitana* ranged from 21.44 to 31.91 pg, while that of *F. yvesii* subsp. *lagascae* was from 13.60 to 22.31 pg. We report the highest ploidy level detected for *Festuca* (2n = 14*x* = 98) and previously unknown cytotypes. A positive correlation between holoploid genome size and chromosome number counts shown herein was confirmed. The morphometric approach showed a slight trend towards an increase in the size of some organs consistent with the variation in the ploidy level. Differences in characters were usually significant only among the most extreme cytotypes of each subspecies, but, even in this case, the high overlapping ranges prevent their distinction.

## 1. Introduction

Polyploidy, or whole-genome duplication (WGD), has long been recognised as a major force in angiosperm evolution [1]. This phenomenon is known to have potential effects on certain life-history traits involving genome structure, gene expression, biochemistry, reproductive development, ecophysiology, competitive ability, growth rates and phenotype [2,3]. The most visible and widespread immediate effect is undoubtedly the increase in cell size [4]. In this respect, several studies have indicated that the polyploidy can induce detectable morphology variation in both microcharacters (e.g., stomata and pollen size) and the dimensions of vegetative (e.g., culm and leaf lengths) and reproductive organs (spikelet, lemma, anther and seed sizes, among others) or in generative features (e.g., flowers number per spikelet or inflorescences per plant) [5,6,7].

Grasses are one of the best models with which to study polyploidy and its effects on genome evolution, as they are more than 80% polyploid or palepolyploid [8]. *Festuca* L. is one of the largest genera of the Poaceae family, with more than 500 perennial species distributed mostly in the temperate regions of the world [9,10]. About 70% of the fescue taxa have undergone at least a whole-genome duplication event during their evolutionary history [11]. A wide range of ploidy levels ranging from diploid (2n = 2*x* = 14) to dodecaploid (2n = 12*x* = 84), with basic chromosome number *x* = 7, have been reported to date [11,12]. The taxonomic difficulties of the genus are well known, including many polyploidy groups with hardly distinguishable species.

*Festuca yvesii* Sennen & Pau (*Festuca* sect. *Festuca*) is a highly variable species comprising several cytotypes (Table 1). According to the most recent taxonomic treatment, *F. yvesii* includes four subspecies widely distributed throughout the Iberian Peninsula [13,14]. The subsp. *yvesii* (2n = 42, 56) and the subsp. *altopyrenaica* (Fuente & Ortúñez) Mart.-Sagarra & Devesa (2n = 28) have an eminently Pyrenean distribution, while the subsp. *summilusitana* (Franco & Rocha Afonso) Mart.-Sagarra & Devesa and the subsp. *lagascae* (Cebolla & Rivas Ponce) Mart.-Sagarra & Devesa, on which this study focuses, are found in the centre, west and north of the Iberian Peninsula. The subsp. *summilusitana* (2n = 42, 70, 84) is highly polymorphic, and its taxonomy has changed in recent years [15,16], with this being closely related to the detection of different chromosome numbers (Table 1), type of substrate and its geographical distribution [12,17]. However, Devesa et al. [13] showed that the diagnostic features employed in order to differentiate the segregated entities are highly variable and are blurred when the overall material is analysed, being practically impossible to assign specimens to an entity without knowing other nontaxonomic data. The subsp. *lagascae* grows in the easternmost part of the Central System and in disjunct populations of northern Spain, and all of them are hexaploids (2n = 42) [17,18]. The octoploid level (2n = 56) has been mentioned exceptionally on the basis of the chromosome counts obtained for the conflictive populations from the Cantabrian Mountains [13,17] (Table 1). This taxon has undergone several changes in its nomenclature, and differs morphologically from the subsp. *summilusitana* by having smaller sizes in certain characters (culm, lemma and panicle, among others). However, the distinction between both subspecies is sometimes difficult owing to the overlap in the range of diagnostic characters, with some transitional populations [13].

Our understanding of ploidy diversity has greatly improved over the last two decades owing to the usage of flow cytometry (FCM). This valuable method makes it possible to not only estimate the genome size (or the nuclear DNA content) in a large number of individuals and populations quickly and accurately but also infer the ploidy levels, irrespective of the number of chromosomes [21,22]. Moreover, the determination of DNA-ploidy level from herbarium and silica-gel-dried material using flow cytometry has been successful for several species, including some fescues [23,24,25,26]. The use of FCM has uncovered an overlooked heterogeneity of cytotypes at an intraspecific level in many genera of vascular plants [22,27,28,29]. It has, therefore, been used as an exploratory tool in studies focused on resolving or identifying taxonomically challenging plant groups [30,31,32,33,34,35,36,37,38]. Data on the amounts of nuclear DNA in plants are growing rapidly [39]. In the genus *Festuca*, several works have satisfactorily evaluated the genome size using FCM [12,40,41,42] and have revealed the occurrence of positive correlations between 2C nuclear DNA contents and chromosome number, at least in closely related species [12]. Some recent studies concerning *Festuca* have also relied on this technique in order to assess whether the size of organs increases with both chromosome number and DNA nuclear content, even within the same species [43,44]. 

Taking into account the complexity of *F. yvesii* in terms of the diversity of cytotypes and phenotypic variability, the aims of the present study were: (1) to estimate the genome size of *F. yvesii* subsp. *summilusitana* and *F. yvesii* subsp. *lagascae* using flow cytometry analysis; (2) to test whether the inferred ploidy levels are correlated with the chromosome counts obtained in this study; (3) to explore the morphological variation of the cytotypes and detect differences between them.

## 2. Materials and Methods

### 2.1. Plant Material and Geographical Origin

Individuals of *F. yvesii* subsp. *summilusitana* and *F. yvesii* subsp. *lagascae* were collected throughout their range (Iberian Peninsula) in order to carry out karyological, genome size and morphological analyses (Figure 1). Field sampling was specifically carried out in several populations from the Sierra da Estrela (S1–S3), the Sierra de Gredos and Candelario (S4–S6), the Sierra de la Demanda and Cebollera (L1–L3), the Sierra de Guadarrama (L4–L7), the Cantabrian Mountains (L8) and in some mountains in the northwest (S7–S9) (Figure 1). The species were identified according to Devesa et al. [13].

Genome size (2C values) and DNA-ploidy levels were estimated by flow cytometry (FCM). A total of 94 individuals from 13 populations were analysed (Figure 1). Fresh leaves were selected from each of the individuals and stored in individual plastic bags at 4 °C for 3–5 days until use. Additionally, seven silica-gel-dried samples obtained from four populations were also analysed as a proxy for the validation of DNA-ploidy levels. With regard to the karyological study, we revisited those conflictive populations in which substantial differences were found between DNA-ploidy level and the number of chromosomes reported in the literature (i.e., populations S4–S6 from the Sierra de Gredos and Candelario and populations L4–L7 from the Sierra de Guadarrama; see Figure 1). The seeds of numerous individuals were collected from these seven selected populations during the fruiting period (late July–August) and stored in paper bags for their later use in chromosome counting and morphometric analyses. More than 200 root tips per population were analysed in order to obtain good metaphases plates. Finally, a total of 135 specimens from these populations were examined to explore the morphological variability between cytotypes. Voucher specimens of the populations studied were deposited in the Herbarium at the University of Cordoba (COFC), Spain.

### 2.2. Flow Cytometric Analyses

Genome size was quantified following the method described by Galbraith et al. [45]. Nuclei were released by chopping approximately 50 mg of fresh leaf material from each *Festuca* individual mixed with 50 mg of an internal reference standard, using a razor blade in a Petri dish containing 1 mL of a nuclear isolation buffer (Woody Plant Buffer, WPB [46]). The nuclear suspension was filtered through a 50 µm nylon filter, after which 50 µg/mL of propidium iodide (PI) and 50 µg/mL of RNAse were added. *Pisum sativum* ‘Ctirad’ (2C = 9.09 pg of DNA [47]) and *Vicia faba* ‘Inovec’ (2C = 26.90 pg of DNA [48]) were used as the internal reference standard for the *F. yvesii* subsp. *summilusitana* and F*. yvesii* subsp. *lagascae* samples, respectively.

The samples were analysed using a CyFlow Space flow cytometer (Sysmex-Partec GmbH., Görlitz, Germany; equipped with a 532 nm green solid-state laser). At least 5000 particles were analysed per sample. Only coefficient of variation values (CV) < 5% were accepted for the 2C peaks in fresh samples and up to 8% in dry material. The holoploid genome size (in picograms) was calculated according to the following formula: (sample G1 peak mean/standard G1 peak mean) × 2C DNA content of standard (pg). DNA-ploidy level was inferred by comparing the sample peaks with those of individuals with a known chromosome number and genome size [21].

### 2.3. Chromosome Counts

Seeds were germinated in Petri dishes on wet filter paper at 20 °C. Seedlings (5–10 days old) were pretreated with 2 mM 8-hydroxyquinoline for 24 h at 4 °C in the dark, fixed in a solution of absolute ethanol and glacial acetic acid (3:1, *v*/*v*) for at least 24 h and stored at 4 °C until use. Root tips were hydrolysed for 2 min in 1N HCl at 60 °C, washed with distilled water and then stained with alcoholic acetic carmine for 24–72 h at room temperature. Finally, stained meristems were squashed in 45% acetic acid [49]. Mitotic metaphases were identified and photographed using light microscopy (Leica DM500).

### 2.4. Morphometric Analyses

Sixteen macrocharacters were chosen and measured when available: culm length, leaf length, leaf diameter, auricle length, spikelet length, number of flowers, upper glume length, lemma length, awn length, anther length, inflorescence length, length of the first node of the inflorescence, number of nodes and branches per inflorescence, first branch length and seed length. Between 2 and 5 measurements were made for each character, with the exception of the seeds. A total of 356 seeds from the populations where the chromosome count was made were measured (35–75 seeds per population). Reproductive and vegetative characters were measured under a 6.3–40× Leica stereomicroscope (model S6 D).

### 2.5. Statistical Analyses

Descriptive statistics of the genome size estimations were calculated for each taxon (mean, maximum and minimum values and standard deviation). After carrying out a normality test (Shapiro-Wilk test), holoploid genome size differences among populations were analysed using a one-way ANOVA with a post hoc Tukey HSD test (*p* < 0.05). The linear regression and Spearman rank correlation between the holoploid genome size (2C values) and chromosome number detected here were performed. Summary statistics of the morphological characters studied were also provided in order to explore variability among cytotypes. Differences among ploidy levels and all 16 character measurements included in our study were analysed by following the same approach as above. In those cases in which the data could not be transformed in order to obtain a normal distribution, nonparametric tests were applied (Kruskal–Wallis followed by Dunn’s post hoc tests, *p* < 0.05; or Wilcoxon–Mann–Whitney test). Correlations between ploidy level and morphological traits were estimated with the Spearman rank correlation. All statistical analyses were carried out using R version 4.0.3.

## 3. Results

### 3.1. Genome Size and DNA-Ploidy Level Variation

Fluorescence histograms of relative nuclear DNA content showed high-quality G_1_ peaks. The coefficients of variation (CVs) for fresh fescue samples were all within the normal range, below 5.0% (mean CVs = 3.48%), indicating reliable estimations (Figure 2A–E and Figure 3A–E). Genome size data for each subspecies and population are shown in Table 2.

Nuclear 2C DNA content has been estimated for the first time in *F. yvesii* subsp. *lagascae*. The holoploid genome size estimates obtained were significantly different (up to 2.35-fold) for the two subspecies studied. The lowest 2C values were observed for *Festuca yvesii* subsp. *lagascae*, ranging from 13.60 to 22.31 pg, while the highest 2C values were found in the *F. yvesii* subsp. *summilusitana* individuals, ranging from 21.44 to 31.91 pg (Table 2, Figure 4). The analysis of variance (ANOVA) revealed significant differences between the 2C values (F_12,81_ = 837.8, *p* < 0.0001) at the subspecies level. Variability in genome size was found among the populations studied, with a maximum variation of 1.33-fold in the subsp. *summilusitana* (S1 vs. S6) and 1.54-fold in the subsp. *lagascae* (L1 vs. L7) (Table 2). Nevertheless, genome size estimates were homogeneous among individuals, with an average variation of 5.9%.

Specifically, *F. yvesii* subsp. *summilusitana* individuals fell within ploidy levels of 10*x* (S1), 12*x* (S2, S3 and S4) and 14*x* (S5 and S6) (Figure 1 and Figure 4A), while individuals of *F. yvesii* subsp. *lagascae* had nuclear DNA contents that fell within ploidy levels of 6*x* (L1 and L2), 8*x* (L3, L4, L5 and L6) and 10*x* (L7) (Figure 1 and Figure 4B). Noticeably, the DNA content of dodecaploids was slightly less than twice that of hexaploids (Table 2).

Variation in DNA content was not continuous and five nonoverlapping DNA-ploidy levels were identified (Figure 4): hexaploids (2C = 13.60–14.46 pg), octoploids (2C = 16.57–18.23 pg), decaploids (2C = 20.51–24.17 pg), dodecaploids (2C = 25.20–27.71 pg) and tetradecaploids (2C = 29.68–31.91 pg).

The silica-gel-dried leaves yielded peaks of sufficient quality to assign DNA-ploidy level, although CVs were often higher (mean CVs = 4.9%). The preliminary estimates of dry individuals from northwestern populations of *F. yvesii* subsp. *summilusitana* (2C = 23.01–25.84 pg; S7, S8 and S9) fell within the same DNA content group as populations inferred as 10*x*, and those of *F. yvesii* subsp. *lagascae* (L8) were within the 8*x* range (Figure 1, Table 2). It should be noted that the dry samples always showed genome size estimates larger than the fresh samples with the same ploidy level, with a difference of up to 1.93 pg for the subsp. *summilusitana* (S1 vs. S9) and up to 2.55 pg for the subsp. *lagascae* (L4 vs. L8) (Table 2).

### 3.2. Chromosome Numbers

A noticeable chromosomal diversity was found for both taxa, and new cytotypes were detected on the basic chromosome number (*x* = 7). The karyological screenings carried out confirmed the ploidy level detected by FCM (Table 2, Figure 2F–H and Figure 3F–H). The Spearman rank correlation showed a strong positive correlation between 2C nuclear DNA content and the chromosome number observed (Spearman’s rho = 0.92; *p* < 0.001) (Figure 5). Unexpectedly high ploidy levels were detected for the F. yvesii subsp. summilusitana populations from the centre-west of the Iberian Peninsula (Table 2): Puerto del Pico population (S4) was dodecaploid (2n = 12*x* = 84; Figure 2F), while Plataforma de Gredos (S5) and La Covatilla (S6) were tetradecaploid (2n = 14*x* = 98; Figure 2G,H), the highest chromosome number reported in the Festuca genus to date.

Two new somatic numbers were found for *F. yvesii* subsp. *lagascae*: 2n = 56 in the individuals from Puerto de Guadarrama (L4), Puerto de Navacerrada (L5; Figure 3F) and Puerto de Morcuera (L6; Figure 3G), and 2n = 70 in those from Puerto de Canencia (L7; Figure 3H), i.e., octoploid and decaploid, respectively (Table 2). In line with the FCM analyses, non-mixed-ploidy populations were detected. A linear regression analysis was performed in order to check the ploidy level of those populations for which no chromosome counts were carried out (*R^2^* = 0.989; *y* = −0.822868 + 0.319799*x*) (Figure 5). The subsp. *summilusitana* populations from the Serra da Estrela, therefore, fit in 10*x* (S1) and 12*x* levels (S2 and S3), while populations of the subsp. *lagascae* from northern Iberian coincide with the 6*x* (L1 and L2) and 8*x* levels (L3) (Table 2).

### 3.3. Morphological Variation

In general, the largest sizes for all the characters were found in the subsp. *summilusitana* (Figure 6), with the exception of the auricle and awn lengths. A detailed summary of the morphological variation observed is provided in Appendix A. In both subspecies, most of the morphological traits increased in size with increasing ploidy; however, no significant differences were observed with respect to all ploidy levels (Figure 6, Appendix A). Moreover, statistical differences between cytotypes were not found for the same characters in both taxa, except for the lemma, upper glume and seed lengths (Figure 6).

The tetradecaploids of *F. yvesii* subsp. *summilusitana* were significantly larger than the decaploids for the upper glume (ranging between 4.00–6.80 mm and 2.60–5.63 mm; Figure 6A), lemma (ranging between 5.60–6.80 mm and 4.50–6.17 mm; Figure 6B) and anther lengths (ranging between 2.50–3.80 mm and 2.20–3.20 mm; Appendix A). The dodecaploids and tetradecaploids had a similar morphological pattern as regards almost all the characters (Figure 6A,B; Appendix A); differences were only found in the length of the seed (Figure 6C). The only character that was significantly different for all the cytotypes was lemma length (*p* < 0.001), although the ranges of variation showed a high overlap (Figure 6B). In the case of *F. yvesii* subsp. *lagascae*, the hexaploids were significantly smaller than the octoploid and decaploid plants as regards the vegetative and reproductive characters studied (with the exception of the awn and anther length, number of flowers per spikelet, number of branches and nodes per inflorescence; Appendix A), while the octoploid and decaploids plants were morphologically indistinguishable (Figure 6D,E). As in the previous species, the differences between both cytotypes were only observed in the length of the seed (Figure 6F). Significant positive Spearman’s correlations between ploidy level and some reproductive traits such as lemma (*r* = 0.62, *p* < 0.001; *r* = 0.56, *p* < 0.001), upper glume (*r* = 0.28, *p* = 0.009; *r* = 0.36, *p* = 0.002), spikelet (*r* = 0.27, *p* = 0.02; *r* = 0.46, *p* < 0.001) and seed lengths (*r* = 0.36, *p* < 0.001; *r* = 0.22, *p* = 0.002) were detected for both the subsp. *summilusitana* and the subsp. *lagascae*, respectively.

The highest coefficients of variation (mean values for each subspecies ranging from 23% to 42%) were observed for several vegetative characters, such as the culm, leaf and auricle lengths, and those related to the panicle (e.g., the number of branches), while the lowest CVs were observed for the spikelet (from 6.72% to 12.51%) and the floral pieces, particularly as regards the lemma length (from 5.08% to 8.50%) (Appendix A).

## 4. Discussion

### 4.1. Cytogenetic Variation

Whole-genome duplication is common in *Festuca*, for which a variation in genome size (2C values) from 3.89 to 25.67 pg has been reported in the literature. These extreme values correspond to the diploid and dodecaploid levels [12,42]. Our FCM data obtained provided clear evidence of unexpectedly high cytogenetic variation in *F. yvesii* subsp. *summilusitana* and *F. yvesii* subsp. *lagascae*. The discrete genome size variation reported in this study is fully correlated with chromosome number counts and corresponds to five ploidy levels (from 2n = 6*x* = 42 to 14*x* = 98). The genome size range (2C values) obtained for each ploidy level is consistent with that estimated previously for phylogenetically related species (data available for up to the dodecaploid level [39]). This supports the FCM is a reliable technique with which to infer the ploidy level in *Festuca*, as other authors have already pointed out [12,24,25,44].

Of the taxa studied, only genome size estimates for *F. yvesii* subsp. *summilusitana* from the Sierra da Estrela populations had been gathered to date. The 2C values obtained for these populations (S1–S3) are compatible with previous data and, according to Loureiro et al. [12], correspond to decaploid (2n = 70 chromosomes; 2C = 22.69 pg) and dodecaploid (2n = 84 chromosomes; 2C = 25.67 pg) levels. The exploratory genome size estimates attained for the northwestern Iberian populations using dry material (S7–S8) coincide with the decaploid level documented by means of chromosome counting [17] and with the increase in genome size registered using dry material with respect to fresh material [50]. However, the hexaploid level previously documented for the centre-west populations (Gredos and the surrounding mountains; S4–S6), which led to the segregation and recognition of these plants as *F. gredensis* [17,18], was not observed in the current study. Contrary to expectations, we detected a ploidy level that was twice as high in the Puerto del Pico population (S4), which corresponded to a dodecaploid level (2n = 84 chromosomes), with genome size estimates similar to those obtained in populations from the Serra da Estrela (S2 and S3). Even more surprising was the very high chromosome number detected in Plataforma de Gredos (S5) and La Covatilla (S6), which correspond with an unknown tetradecaploid level (2n = 98 chromosomes). This is the first record of such a high level of ploidy for the whole genus and, therefore, also the largest genome size recorded (average 2C values ranging from 30.31 to 30.48 pg). These data suggest that high ploidy levels might be more represented in *Festuca* sect. *Festuca* species than previously assumed. It is interesting to note that the mean DNA content increased with ploidy level, but not proportionally. The mean genome size obtained in the dodecaploids (S2, S3 and S4) was, therefore, slightly lower than the theoretical double mean DNA content of related hexaploids (L5 and L6). This decrease in the monoploid genome size (i.e., in the DNA content divided by the ploidy level) observed at the highest ploidy levels would appear to be a general trend in angiosperms. Genome downsizing with polyploidy has been reported in *Festuca* and other Poaceae genera [12,27,51,52] and could result from genomic changes focused on reducing the negative effect of increased DNA content [53].

Although a mosaic of ploidy levels was found throughout the distribution area of *F. yvesii* subsp. *summilusitana*, the variation among individuals was usually low in the majority of populations. Notwithstanding, in the absence of changes in ploidy, some within-population variations in genome size (up to 12%) could be attributed to the presence of B-chromosomes and/or to a variation in the amount of noncoding DNA and in the activity of transposable elements [54,55]. Both phenomenons are relatively common in Poaceae and have been repeatedly documented for this taxon [17] and other fescues species such as *F. rupicola* Heuff., *F. vaginata* Willd., *F. polesica* Zapal. and *F. pallens* Host [24,54].

The genome size estimations obtained for *F. yvesii* subsp. *lagascae* are provided here for the first time. The predominant hexaploid level previously reported for this taxon [17,18] partially coincided with our results. The FCM measurements indicated that only in two northern populations (Valdezcaray and Trigaza, L1 and L2) did the genome size values fit within the expected range for the hexaploid species of *Festuca* sect. *Festuca* (2C values varying from 12.30 to 14.22 pg) [42]. By contrast, DNA content in the remaining populations coincided with that obtained in other related octoploid species such as *F. brigantina* (2C = 17.08 pg) [12] or *F. laevigata* (2C = 18.60 pg) [42], or it was similar to that found in decaploid populations of the subsp. *summilusitana*. Both levels are new in the Sierra de Guadarrama populations, where only hexaploids had been detected previously [17]. Our approach with dry material for the most northwestern population of this subspecies (Puerto de San Glorio, L8) fits with the octoploid level previously reported for nearby populations [17]. 

Although the presence of more than one cytotype has also been reported for several Iberian fescues species (e.g., *F. ampla*, 2n = 4*x* and 6*x*; *F. iberica*, 2n = 4*x* and 6*x*) or even in subspecies (e.g., *F. vasconcensis* subsp. *vasconcensis*, 2n = 6*x* and 8*x*, *F. rubra* subsp. *juncea* 2n = 6*x* and 8*x*) [12,13,17], none of them appear to have reached the variation found for the *F. yvesii* subspecies. With regard to the *Festuca* sect. *Festuca* species, the episodes of genome duplication have been suggested to operate in independent evolutionary processes within each mountain range, most probably originating from one common ancestor. However, the current data make it very difficult to discern whether the cytological variability observed in nature results from auto- or allopolyploidisation processes, despite the fact that the latter is believed to have played a major role in the evolution of the genus [56]. The divergence and radiation of the *Festuca* sect. *Festuca* taxa occurred relatively recently, in the Pleistocene (c. 1.9 Ma) [57], coinciding with the colonisation of mountain systems in the Iberian Peninsula during glaciation periods [58,59]. In this context, genome duplications could have constituted a success mechanism of adaptation and range expansion, along with speciation, in changing climatic conditions [8,60,61,62]. Phylogenetic analyses based on nuclear and chloroplast markers have detected polytomies or weakly supported relationships among *Festuca yvesii* subspecies [63]. This unresolved phylogeny may be substantially influenced by the intricate history of past and recent hybridisation and polyploidisation [56].

### 4.2. Morphological Variation

Genome duplications have, owing to their effects on cell size and gene expression, been postulated to mediate morphological and physiological shifts in new cytogenetic entities [8]. Moreover, genome size variation may itself be a source of phenotypic instability [64]. *Festuca yvesii* is a polyploid complex with low morphoanatomical differentiation across subspecies. Despite the heterogeneity in the morphological traits and the high range of overlap found among the cytotypes of each taxon, interesting morphological trends appear with regard to the ploidy-level variation detected here. 

According to these results, the lemma, upper glume and seed lengths in particular responded significantly to changes in ploidy in both subspecies, similar to what was observed in other taxa [43,44]. These traits also had low phenotypic plasticity, which would indicate that they are genetically fixed. We expected to find the same pattern for the anther and spikelet lengths in both subspecies, but differences for the anther length were detected only between cytotypes of *F. yvesii* subsp. *summilusitana*, while differences in spikelets were observed only in the subsp. *lagascae*. This could respond to an insufficient number of anthers measured in the subsp. *lagascae* owing to the state of conservation of the material. In the case of the spikelet, the variations in the spikelets are related to the number of flowers of which they are formed. Despite the above, the differences observed in the arithmetic mean obtained for most of the characters are significant only among the most extreme cytotypes of each subspecies, and even in these cases, the ranges of variation overlap. In fact, these cytotypes connect morphologically in such a way that no clear lines of distinction can be drawn between them. In this sense, the relationship between cytotype and the size of morphological features does not necessarily have to be allometric and is often subject to intricate genetic and epigenetic regulatory networks [2,65]. As occurred with our results, Rewicz et al. [43] found differences between the organ size of the diploid and tetraploid of *F. amethystina* (sect. *Aulaxyper*), although these were not sufficient to clearly distinguish both cytotypes on the basis of the measurements themselves. In contrast, most of the vegetative characters (e.g., culm and leaf), along with the panicle length and the traits associated with them, were highly variable and could be closely related to adaptation to certain abiotic conditions (e.g., the availability of nutrients, exposure and drought) and/or biotic interactions (e.g., herbivory). The differences found among the cytotypes of the subsp. *lagascae* as regards culm, leaf and panicle lengths should, therefore, be examined under the same garden growing conditions. Several studies focused on unravelling the taxonomic relationships among cytotypes have experienced difficulties in separating them morphologically and recognising them as entities owing to the lack of reliable diagnostic characters, their heterogeneity and the overlap in the range of variation, especially in groups with complex chromosomal diversity [13,34,66], as has also been detected in the subspecies studied.

## 5. Conclusions

Our data provide new insights into the cytogenetic complexity of *F. yvesii* regarding the great variation in genome size and chromosome number diversity. Flow cytometric analyses complemented with chromosome counts suggest an intricate polyploid series at the subspecies level, with cytotypes not previously detected. We have additionally found the highest chromosome number (2n = 14*x* = 98) and the largest genome size estimates reported for the genus. Likewise, our morphological data, although exploratory, revealed a certain trend in the morphological pattern related to cytogenetic variation, especially in the lemma, but this was clearly insufficient for the morphological distinction of the cytotypes owing to their wide overlap. This study opens up the possibility of further research with which to explore cytogenetic diversity in fescue species with complex morphological pattern variation, by considering the advantages of the flow cytometric method in the ploidy check at the population level.

## Figures and Tables

**Figure 1 genes-12-00906-f001:**
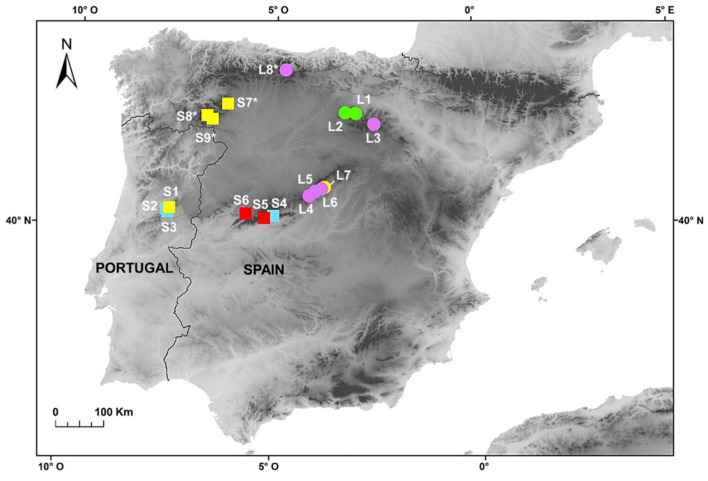
Populations of *F. yvesii* subsp. *summilusitana* (squares) and *F. yvesii* subsp. *lagascae* (circles) sampled in the Iberian Peninsula. Ploidy-level variation is shown using coloured symbols: purple, 6*x*; green, 8*x*; yellow, 10*x*; light blue, 12*x*; and red, 14*x*. The asterisks (*) indicate populations estimated from dried material.

**Figure 2 genes-12-00906-f002:**
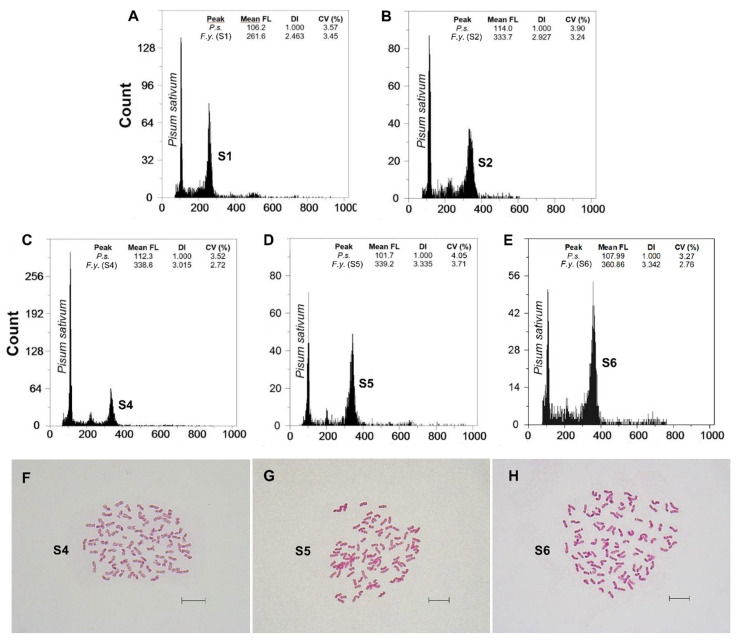
(**A**–**E**) Flow cytometric histograms of relative PI fluorescence intensities and (**F**–**H**) mitotic metaphase chromosome preparations (for the S4, S5 and S6 populations) of *F. yvesii* subsp. *summilusitana*. Abbreviations: S1, ManTable S2. Sabugueiro; S4, Puerto del Pico (2n  =  12*x*  =  84); S5, Plataforma de Gredos (2n  =  14*x*  =  98); and S6, La Covatilla (2n  =  14*x*  =  98). Mean FL, mean relative fluorescence in picograms; DI, DNA index; CV (%), coefficient of variation of the peak in percent. Scale bar: 10 μm.

**Figure 3 genes-12-00906-f003:**
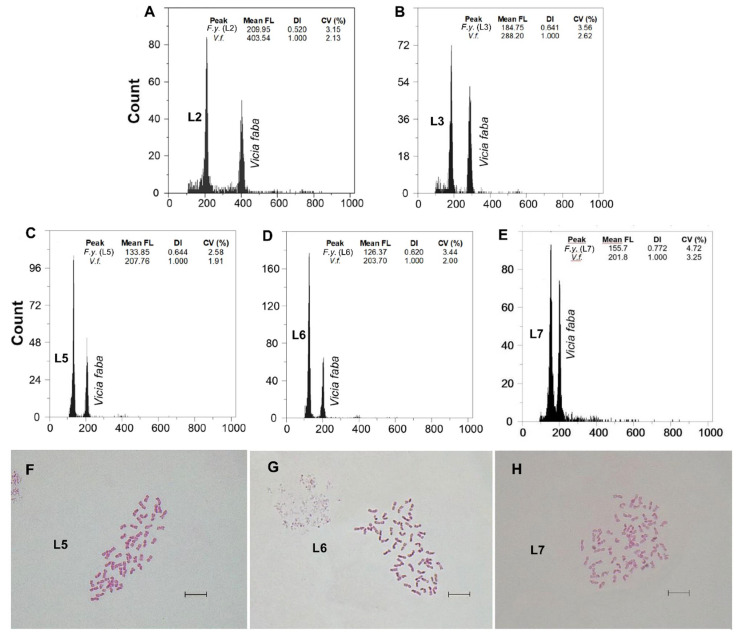
(**A**–**E**) Flow cytometric histograms of relative PI fluorescence intensities and (**F**–**H**) mitotic metaphase chromosome preparations (for the L5, L6 and L7 populations) of *F. yvesii* subsp. *lagascae*. Abbreviations: L2, Trigaza; L3, Puerto de Piqueras; L5, Puerto de Navacerrada (2n  =  8*x*  =  56), L6, Puerto de la Morcuera (2n  =  8*x*  =  56); and L7, Puerto de Canencia (2n  =  10*x*  =  70). Mean FL, mean relative fluorescence in picograms; DI, DNA index; CV (%), coefficient of variation of the peak in percent. Scale bar: 10 μm.

**Figure 4 genes-12-00906-f004:**
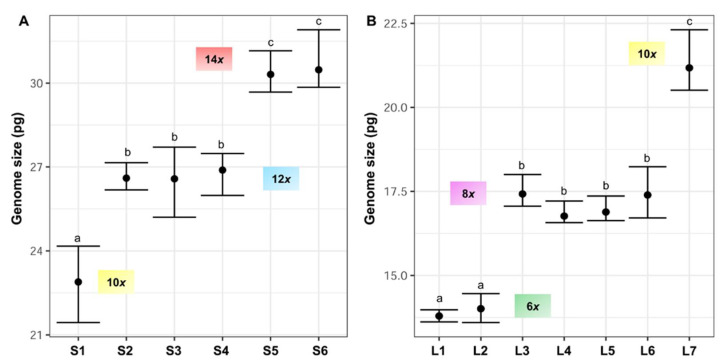
Holoploid genome size range and mean (2C values, in picograms) in the sampled populations of (**A**) *F. yvesii* subsp. *summilusitana* (S1–S6) and (**B**) *F. yvesii* subsp. *lagascae* (L1–L7). DNA-ploidy levels are marked (6*x*, 8*x*, 10*x*, 12*x* and 14*x*). Only estimates from fresh material are represented. Abbreviations (S1–S6 and L1–L7) correspond to the populations indicated in Table 2. Significant differences among populations at *p* < 0.05 are denoted by different letters.

**Figure 5 genes-12-00906-f005:**
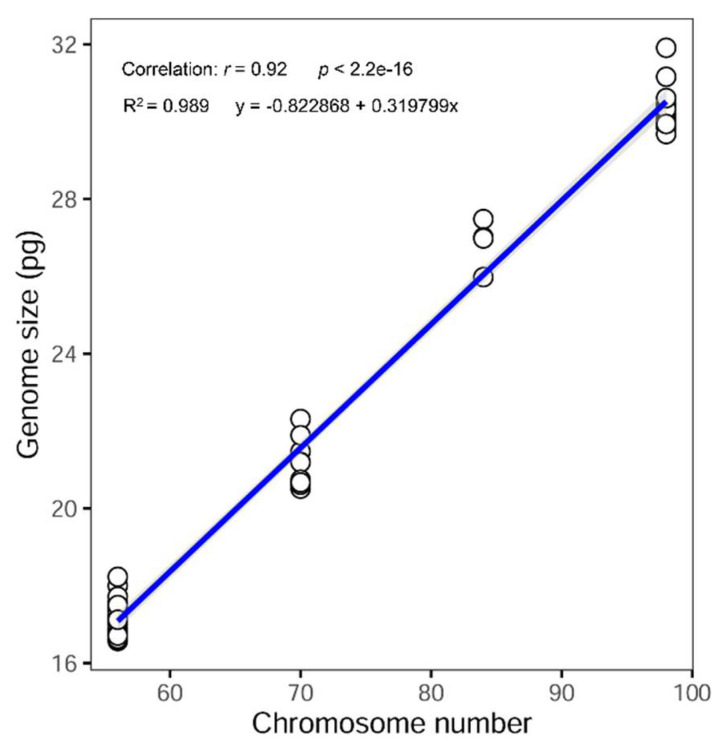
Linear regression between holoploid genome size estimates (2C values, in picograms) and chromosome number observed in this study (2n = 56, 70, 84, 98), including 95% confidence intervals. The linear regression equation and coefficient are provided. The correlation is statistically significant (*p* < 0.0001).

**Figure 6 genes-12-00906-f006:**
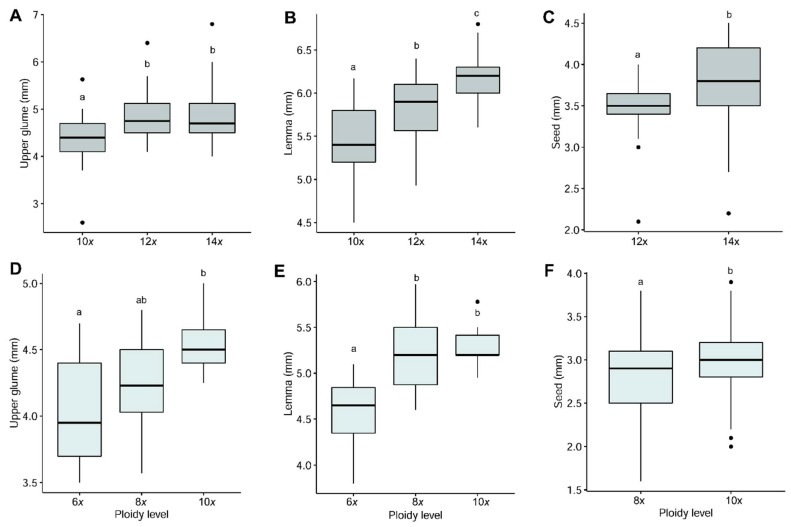
Box plot indicating variation in the lemma, upper glume and seed lengths (in millimetres) in the different ploidy levels detected in (**A**–**C**) *Festuca yvesii* subsp. *summilusitana* and (**D**–**F**) *F. yvesii* subsp. *lagascae*. Outliers are shown as dots. Significant differences among cytotypes at *p* < 0.05 are denoted by different letters.

**Table 1 genes-12-00906-t001:** Ploidy level and chromosome number reported in the literature for subspecies of *F. yvesii* by chromosome counting or FCM analysis. The reference column provides the original source of publication to each chromosome count. Superscript indicates the taxon name it was published under. The asterisk (*) indicates chromosome counts in extra-Iberian material.

Subspecies (According to Devesa et al., [13])	2n	Method	References
*F. yvesii* subsp. *yvesii*	2n = 6*x* = 42 *	Counting ^1^	[19]
	2n = 8*x* = 56	Counting ^1^	[17]
*F. yvesii* subsp. *altopyrenaica*	2n = 4*x* = 28	Counting ^2^	[20]
*F. yvesii* subsp. *summilusitana*	2n = 6*x* = 42	Counting ^3,4^	[17,18]
	2n = 10*x* = 70	Counting and FCM ^4^	[12,17]
	2n = 12*x* = 84	Counting and FCM ^4^	[12]
*F. yvesii* subsp. *lagascae*	2n = 6*x* = 42	Counting ^5^	[17]
	2n = 8*x* = 56	Counting ^1^	[17]

^1^ *F. yvesii*, ^2^ *F. altopyrenaica*, ^3^ *F. gredensis*, ^4^ *F. summilusitana*, ^5^ *F. curvifolia*.

**Table 2 genes-12-00906-t002:** Holoploid genome size estimations (mean, standard deviation of the mean and maximum and minimum values) for the sampled populations of each subspecies. Data regarding the population and number of individuals sampled (*n*) are provided. The last column shows the chromosome counts carried out in this study. Type of material analysed: F, fresh; D, dry.

Taxon	Mountain Range	Locality	n	2C Mean ± SD (pg)	2C Range (pg)	Material	DNA-Ploidy Level	2n
***F. yvesii* subsp. *summilusitana***	W range: Serra da Estrela (Portugal)	S1: Castelo Branco, Manteigas	9	22.89 ± 0.96	21.44–24.17	F	10*x*	-
		S2: Castelo Branco, Sabugueiro	8	26.60 ± 0.35	26.18–27.15	F	12*x*	-
		S3: Castelo Branco, Torre	6	26.58 ± 0.87	25.20–27.71	F	12*x*	-
	C-W range: Sierra de Gredos and Candelario (Spain)	S4: Ávila, Puerto del Pico	5	26.89 ± 0.55	25.98–27.48	F	12*x*	84
		S5: Ávila, Plataforma de Gredos	7	30.31 ± 0.49	29.68–31.16	F	14*x*	98
		S6: Salamanca, La Covatilla	6	30.48 ± 0.73	29.85–31.91	F	14*x*	98
	NW mountains (Spain)	S7: León, Villalibre de Somoza	3	24.67 ± 1.47	23.01–25.84	D	10*x*	-
		S8: Zamora, Sierra Segundera	1	24.11		D	10*x*	-
		S9: Zamora, Alto de San Juan	1	24.82		D	10*x*	-
***F. yvesii* subsp. *lagascae***	*n* range: Sierra de la Demanda and Cebollera (Spain)	L1: Logroño, Valdezcaray	6	13.79 ± 0.15	13.62–13.98	F	6*x*	-
		L2: Burgos, Trigaza	10	14.01 ± 0.32	13.60–14.46	F	6*x*	-
		L3: Logroño/Soria, Puerto de Piqueras	10	17.42 ± 0.29	17.06–18.00	F	8*x*	-
	C range: Sierra de Guadarrama (Spain)	L4: Madrid, Puerto de Guadarrama	7	16.76 ± 0.22	16.57–17.21	F	8*x*	56
		L5: Madrid, Puerto de Navacerrada	7	16.88 ± 0.24	16.63–17.36	F	8*x*	56
		L6: Madrid, Puerto de la Morcuera	4	17.39 ± 0.65	16.71–18.23	F	8*x*	56
		L7: Madrid, Puerto de Canencia	9	21.18 ± 0.62	20.51–22.31	F	10*x*	70
	North range: Cantabrian mountains (Spain)	L8: Puerto de San Glorio	2	19.31 ± 0.41	19.02–19.59	D	8*x*	-

## Data Availability

All data generated or analysed during this study are available from the corresponding author on reasonable request.

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
