# Peer review of "Genome Size, Chromosome Number and Morphological Data Reveal Unexpected Infraspecific Variability in *Festuca* (Poaceae)"

_genes, 2021, doi:10.3390/genes12060906_

Round 1
Reviewer 1 Report
The manuscript of Martinez-Sagarra et al. entitled ‚Genome size, chromosome number and morphological data reveal unexepected infraspecific variability in Festuca (Poaceae)‘ describes the cytogenetic variation of 17 populations of two subspecies of Festuca yvesii. The authors performed flow cytometric genome size estimations, chromosome counts and compared them to certain morphological traits. The experiments are well peformed, nicely presented and thoroughly discussed.
I only have some minor issues:
For the flow cytometric analysis the authors wrote that they only accepted measurements with an CV below 5% (M& M). However, in the results section they reported that the CVs were all within the normal range, below 4 % although Fig. 3E shows a histogram with a CV of 4.72. This contradiction should be clarified.
For Fig 3 I would have preferred to see the positioning of the peaks in a comparable manner, e.g. the reference peak of Vicia always on channel 400 (like in Fig 2 where the Pisum peak is always at channel 100). This would allow an easier estimation of the ploidy levels for the readers. However, since the obtained data are not influenced by it there is no need to perform extra measurements. Additionally, I would recommend for further analysis to define the threshold not so close to the first peak.
The authors explain the within-population variation in genome size (up to 12%) with the potential occurrence of B-chromosomes. Are B-chromosomes described for F. yvesii? Were indications for B-chromosomes found during the chromosome counts? Unfortunately the material with the highest variation was not analysed cytologically.
‘infraespecific’ should be corrected to ‘infraspecific’ in the title
Author Response
Reply to reviewer 1
1) For the flow cytometric analysis the authors wrote that they only accepted measurements with an CV below 5% (M& M). However, in the results section they reported that the CVs were all within the normal range, below 4 % although Fig. 3E shows a histogram with a CV of 4.72. This contradiction should be clarified.
Response: It was a typographic error. We accepted CVs below 5%.
2) For Fig 3 I would have preferred to see the positioning of the peaks in a comparable manner, e.g. the reference peak of Vicia always on channel 400 (like in Fig 2 where the Pisum peak is always at channel 100). This would allow an easier estimation of the ploidy levels for the readers. However, since the obtained data are not influenced by it there is no need to perform extra measurements. Additionally, I would recommend for further analysis to define the threshold not so close to the first peak.
Response: We completely agree with the reviewer and will use this best practice in future works. Many thanks for the advice.
3) The authors explain the within-population variation in genome size (up to 12%) with the potential occurrence of B-chromosomes. Are B-chromosomes described for F. yvesii? Were indications for B-chromosomes found during the chromosome counts? Unfortunately the material with the highest variation was not analysed cytologically.
Response: We did not find B-chromosomes in the populations studied of F. yvesii. However, we cannot rule out it, especially in populations with a high level of ploidy, in which chromosome counting is difficult.
4) ‘infraespecific’ should be corrected to ‘infraspecific’ in the title
Response: Done.
Reviewer 2 Report
The aims of the present study are 1) to estimate the genome size in a number of Festuca yvesii subsp. summilusitana and F. yvesii subsp. lagascae populations by means of FCM, 2) to test the correlation between chromosome counts and genome size (or DNA ploidy in dried samples) among them, 3) to explore the morphological variation in their cytotypes and test correlation to ploidy.
Although, the study is already at an advanced state, it exhibits minor problems, which are as follows:
Abstract: “A positive correlation between genome size and chromosome number counts shown herein was confirmed.” better write “holoploid genome size”.
Why did you use a woody plant buffer, although Festuca is not a woody plant?
Why is the name of the buffer used “woody plant nuclear” and not “woody plant buffer” as given in the cited reference? Is this WPN a modified version of WPB? If yes, give the kind of modification.
The name of the flow cytometer producer is Sysmex-Partec.
The peak CVs are rather high, i.e. up to 5.65% especially in the P.s. peaks! Pisum sativum usually exhibits very good performance and very low CVs upon FCM, therefore it was generally accepted to be used as a standard species. In a previous work dealing with other Festuca sp. (12), a different buffer was used, which resulted in low CV% in the Festuca samples as well as in P.s. Why was the WPN buffer chosen for the present work, although the buffer used in (12) showed much better CVs?
Figure 2, L7 contradicts the main text: “The coefficients of variation (CVs) for fresh fescue samples were all within the normal range, below 4.0% (mean CVs 3.48%), indicating reliable estimations (Figures 2A-E 189 and 3A-E).” Figure L7 shows a Festuca peak (fresh sample) with 4.72% CV.
In the M&Ms, you stated to have excluded measurements with peak CVs higher than 5%. Why were the measurements as shown in figure 2D and E accepted? Also the standard peak quality impacts on the reliability of the result, hence should exhibit low CV%.
In order to distinguish real genome size variation from measurement errors within a ploidy level, it is strongly recommended to show histograms of highest and lowest ranking samples prepared together (without a standard organism) - one histogram per ploidy level.
Page 8: “…and new cytotypes were detected on the basis of the basic number (x=7).” better write “…basic chromosome number…”.
Improve the terminology within the whole manuscript in regard of the nuclear DNA amounts. E.g. page 11 “The decrease in DNA in a higher ploidy would appear to be a general trend in angiosperms;…”. Is the DNA really decreasing at higher ploidy levels in your study objects? The table contradicts, as it shows increasing nuclear DNA amounts positively correlated with the ploidy levels! In fact, the manuscript lacks analysis in regard of the monoploid genome size (Cx-value) in the Festuca populations studied. If you feel unsure in regard of terminology, see Greilhuber et al. 2005, Annals of Botany 95: 255- and Greilhuber et al. 2009, Chromosoma 118: 391-.
S1 Table: replace “Media” by “Mean” in the header of the table, if the mean is given, and “Median”, if the median is given.
Insert a thin line between the subspecies, or give the subspecies name in every first line.
Author Response
Reply to reviewer 2
1) Abstract: “A positive correlation between genome size and chromosome number counts shown herein was confirmed.” better write “holoploid genome size”.
Response: Done.
2) Why did you use a woody plant buffer, although Festuca is not a woody plant?
Response: Despite the buffer is called woody plant buffer (WPB), it proved to be useful for a wide range of species, woody or not. Also, despite Festuca is not a woody plant the rigidness of its leaves makes it a slightly troublesome species to isolate nuclei and analyse.
3) Why is the name of the buffer used “woody plant nuclear” and not “woody plant buffer” as given in the cited reference? Is this WPN a modified version of WPB? If yes, give the kind of modification.
Response: It was a typographic error. Woody plant buffer (WPB) is the correct of the buffer, and so it was modified in the manuscript.
4) The name of the flow cytometer producer is Sysmex-Partec.
Response: Done.
5) The peak CVs are rather high, i.e. up to 5.65% especially in the P.s. peaks! Pisum sativum usually exhibits very good performance and very low CVs upon FCM, therefore it was generally accepted to be used as a standard species. In a previous work dealing with other Festuca sp. (12), a different buffer was used, which resulted in low CV% in the Festuca samples as well as in P.s. Why was the WPN buffer chosen for the present work, although the buffer used in (12) showed much better CVs?
Response: As it can be seen by the histograms provided, in general, the quality of the analyses was high, with low CVs and low amount of background debris. Usually, in general, CV is increasing towards the left side (similar variance but lower peak mean) of the scale, and this may have influenced the CVs of Pisum sativum, who were set in the 100 position of the 1024 scale. Still, we double checked the two cases where the CV of Pisum sativum was above 5% and it was typographic error. If we compare the quality of the peaks of Figure 2D and 2E, they are similar to that of Figures 1A, B and C. This was corrected.
WPB is an improved version of the Tris.MgCl2 buffer used in (12), as it uses some of the components of this buffer together with components of LB01 buffer.
6) Figure 2, L7 contradicts the main text: “The coefficients of variation (CVs) for fresh fescue samples were all within the normal range, below 4.0% (mean CVs 3.48%), indicating reliable estimations (Figures 2A-E 189 and 3A-E).” Figure L7 shows a Festuca peak (fresh sample) with 4.72% CV.
Response: This error was modified in the body text: range below 5% was accepted to fresh samples of Festuca.
7) In the M&Ms, you stated to have excluded measurements with peak CVs higher than 5%. Why were the measurements as shown in figure 2D and E accepted? Also the standard peak quality impacts on the reliability of the result, hence should exhibit low CV%.
Response: As referred above the CVs were double checked and the correct CVs of Figure 2D and E are 4.05% and 3.27%. The Figure 2 (values of P. sativum in Figure 2D and 2E) has been modified accordingly.
8) In order to distinguish real genome size variation from measurement errors within a ploidy level, it is strongly recommended to show histograms of highest and lowest ranking samples prepared together (without a standard organism) - one histogram per ploidy level.
Response: This is a great suggestion and best practice made by the reviewer and as we do not have fresh material to do such analyses, we will consider it in future studies.
9) Page 8: “…and new cytotypes were detected on the basis of the basic number (x=7).” better write “…basic chromosome number…”.
Response: Done.
10) Improve the terminology within the whole manuscript in regard of the nuclear DNA amounts. E.g. page 11 “The decrease in DNA in a higher ploidy would appear to be a general trend in angiosperms;…”. Is the DNA really decreasing at higher ploidy levels in your study objects? The table contradicts, as it shows increasing nuclear DNA amounts positively correlated with the ploidy levels! In fact, the manuscript lacks analysis in regard of the monoploid genome size (Cx-value) in the Festuca populations studied. If you feel unsure in regard of terminology, see Greilhuber et al. 2005, Annals of Botany 95: 255- and Greilhuber et al. 2009, Chromosoma 118: 391-.
Response: We must apologize for the misconception due to an inaccurate terminology in this sentence. It has been re-phrased following Greilhuber (2009), and we carefully reviewed the whole manuscript to solve such inconsistencies. Considering the length of the manuscript we decided to leave the analysis of monoploid genome size variation aside. However, if the reviewer insists we can do it in the future.
11) S1 Table: replace “Media” by “Mean” in the header of the table, if the mean is given, and “Median”, if the median is given.
Response: Done.
12) Insert a thin line between the subspecies, or give the subspecies name in every first line
Response: Subspecies names were inserted in every line in S1 Table.